

# Effects of pH, temperature and hydraulic disturbance on nitrogen release from sediments in the Sunxi River, Three Gorges Reservoir Area, China

Yihong Ning[1], Bin Gao[2], Haiyan Wang[1] and Wenning Hou[1]

[1] State Key Laboratory of Efficient Production of Forest Resources, Key Laboratory of Forest Cultivation and Protection, Ministry of Education, College of Forestry, Beijing Forestry University, Beijing, China
[2] College of Smart Urban Construction, Guangzhou City Polytechnic, Guangzhou, China

## ABSTRACT

To clarify the influence of changes in the overlying water environment on internal nitrogen release from reservoir sediments, we collected surface sediments at a depth of approximately 10 cm from the Sunxi River in the tail area of the Three Gorges Reservoir area for simulation experiments. By using orthogonal simulation experiments in the laboratory, we studied the effects of water pH, temperature and hydraulic disturbance on nitrogen release in the sediment and established a quantitative linear relationship between the nitrogen release rate from the sediment and the environmental factors of the overlying water. The results indicated that the average concentrations of total nitrogen (TN) and total phosphorus (TP) in the sediment were 430 mg/kg and 200 mg/kg, respectively. The sediment TN concentration had a very significant positive correlation with the sediment organic matter content ($P < 0.001$). The sediment TN, $NO_3$-N and $NH_4$-N release intensities gradually increased with increasing incubation time, with maximum release rates of 29.24 mg/((m²·d), 23.11 mg/(m²·d) and 4.32 mg/((m²·d), respectively. Range analysis revealed that the significance of the effects of environmental factors on sediment TN and $NH_4$-N release were ranked as follows: temperature > pH > disturbance, and that of $NO_3$-N release was ranked as pH > temperature > disturbance. Temperature plays the most important role in the behavior of different forms of nitrogen release from sediments. The capacity and potential for nitrogen release from sediments offer crucial insights for assessing the risks posed to the overlying water and highlighting the importance of these factors in water quality management and prediction in the reservoir area.

# INTRODUCTION

Surface water, an essential component of ecosystems such as wetlands, rivers, and lakes, plays a critical role in maintaining ecological balance and biodiversity and supporting agriculture, industry, and tourism (*Albou et al., 2024*). Dissolved and particulate non-point source pollutants enter receiving water bodies through rainfall runoff or surface flow

Corresponding authors
Bin Gao, 2368469815@qq.com
Haiyan Wang,
haiyanwang72@aliyun.com

from diffuse sources. If these pollutants are not promptly removed or treated, they can accumulate in the riverbed, leading to the formation of endogenous pollutants in sediments (*Jaskula et al., 2021*; *Mcglue et al., 2021*). Sediments are mixtures of particulate matter, organic material, and chemical substances deposited on the bottom of natural waters and constitute a key part of aquatic ecosystems (*Cheng et al., 2018*; *Du et al., 2022*). Nitrogen (N) has received attention as an indicator of water quality and non-point source pollution. N in water bodies and sediments predominantly exists as organic N, with inorganic N being the next most common form. Sediments can act as a potential source of ammonium to overlying water through the process of ammonification of organic N (*Fan et al., 2019*). When organic pollutants accumulate in large quantities, their mineralization and decomposition release dissolved N into sediment pore water (*Pulley & Collins, 2024*). N release from sediments can significantly degrade water quality and contribute to ongoing eutrophication, particularly when external nutrient sources are effectively controlled (*Danyang et al., 2021*; *Han et al., 2020*). A study on the sediments of Wisconsin Lake revealed significant denitrification and N immobilization through laboratory and field experiments using submerged wide-mouth plastic bottles (*Chen et al., 1972*). Sediment is therefore referred to as a sink for nutrients in the aquatic environment. However, under certain conditions, it can act as a nutrient source. Disturbances such as dredging, strong currents, storms, or changes in water chemistry can remobilize N into the water column (*Eggleton & Thomas, 2004*; *He et al., 2015*). Environmental factors, including pH, temperature, dissolved oxygen (DO), salinity, and microbial activity, also play key roles in triggering N release, leading to water quality degradation and endogenous pollution (*Cornwell et al., 2016*; *Zhang et al., 2017*). In rivers with severe sediment pollution, the contribution of endogenous pollution to overlying water can rival or even surpass exogenous inputs (*Aldrees et al., 2022*; *Peng et al., 2021*).

N in sediments can be released into water bodies through biological, physical and chemical mechanisms (*Kumwimba et al., 2023*). This process typically involves N entering interstitial water and then diffusing upwards under the influence of environmental factors such as temperature, dissolved oxygen (DO), pH, and microbial activity (*Liu et al., 2021*). pH affects oxidation–reduction, absorption–desorption, and precipitation–solubilization reactions by altering the chemical, biological and aquatic reactions of nutrients (*Zhang et al., 2015*). Under acidic or alkaline conditions, elements can more readily gain or lose electrons and thus alter their reactivity and availability. This phenomenon is particularly important in the processes of nutrient cycling and pollution degradation. In addition, pH determines the solubility of certain compounds, causing nutrients to either precipitate and become unavailable or remain dissolved and bioavailable (*Hong, Kinney & Reible, 2011*).

Temperature affects the solubility of carbonate and hydroxide minerals in sediments and regulates the metabolic rates of microorganisms and aquatic plants, thereby influencing the decomposition of organic matter and the mobility of nutrients (*Robador et al., 2010*). Studies in the Pearl River Delta revealed that temperature variations significantly influence the process of N release from sediments, particularly ammonium and nitrate concentrations (*Zhu et al., 2023*). Elevated temperatures accelerate chemical reactions, enhancing N release into the water column and intensifying eutrophication risk during warmer seasons.

Hydraulic disturbances significantly influence pollutant release from sediments by mobilizing contaminants previously bound in sediment layers (*Ma et al., 2023*). Natural events such as storms and human activities such as dredging can resuspend sediments, redistributing both organic and inorganic pollutants into the water column. An increased water velocity generates turbulence, accelerating nutrient release from surface sediments (*Castaldelli et al., 2018*). Additionally, hydraulic disturbances alter sediment transport and deposition patterns, spreading contaminants across wider areas. Studies in shallow lakes have shown that strong hydrodynamic disturbances notably increase N release, particularly when sediments are resuspended in the overlying water. This suggests that managing hydrodynamic conditions in reservoirs is crucial for controlling N pollution (*Nie et al., 2024*).

Many studies have explored the relationships between environmental factors and pollutant release, with changes in nutrient salts providing insights into their release patterns in sediments. *Gong, Liu & Yang (2021)* investigated nitrogen (N) and phosphorus (P) release from rural ditch sediments on the northeastern bank of Dian Lake, Yunnan Province, China. Through indoor static incubation experiments, they analysed nutrient dynamics under various pH conditions. The results revealed that alkaline and acidic conditions significantly increased the release of N and P in rural ditch sediments. *Lei et al. (2016)* investigated the effects of environmental factors, including disturbance, pH, and DO, on the release of COD, TP, TN, and $NO_3$-N. They reported that TP and TN concentrations increased with increasing disturbance intensity and decreasing DO levels, whereas a pH range of 6.0−8.0 was most effective at inhibiting pollutant release. On the basis of field measurements and laboratory experiments, *Nguyen et al. (2019)* investigated the adsorption/desorption capacity of P onto suspended sediments along the salinity gradient in the Saigon River estuary, southern Vietnam. *Lee & Oh (2018)* investigated the effects of sediment release on water quality and pollution in four agricultural reservoirs in South Korea and reported that the release concentrations of $NO_3$-N increased under oxic conditions and those of $NH_4$-N and TP increased under anoxic conditions. To explore the synergistic effects between factors, complex experimental designs, such as orthogonal experiments and response surface tests were applied to simulate the release patterns of sediments under the combined effects of several environmental factors and to determine the significance ranking of each factor. *Zhang et al. (2017)* applied the $L_{27}(3^{13})$ orthogonal design to analyse the effects of four factors, namely, sediment type, temperature, DO and pH, and their interactions on nutrient exchange fluxes at the sediment-water interface, revealing primary and secondary relationships among these factors. Similarly, *Lu et al. (2022)* used orthogonal simulation experiments in the Dahekou Reservoir, Inner Mongolia, to establish a quantitative relationship between N release intensity and environmental factors.

The Sunxi River, located within the Three Gorges Reservoir area, extends 120 km and crosses Jiangjin city, Chongqing, China. It is characterized by its long river course, significant elevation drop, and extensive coverage area. N was identified as the primary pollutant in this river (*Hou et al., 2022*). Excessive N in the water can easily transfer to and accumulate in the sediments, making the sediments a potential new source of N.

Currently, most studies on sediments focus on sampling at one or a few points within a river or reservoir, with few studies conducting full-scale sampling across the entire river basin. Therefore, we collected sediment samples from 34 evenly distributed points along the Sunxi River for the determination of nutrient concentrations and spatial distribution patterns. Indoor simulation experiments were conducted to investigate the relationships between sediment N release and environmental factors such as disturbance level, temperature, and pH. By calculating the release rate and establishing a regression equation, this study aims to elucidate the mechanisms of N release from sediments under different environmental conditions. The main objective of this study was to establish a comprehensive understanding of how the overlying water's environmental factors influence N release rates, revealing the processes that drive N dynamics in the Sunxi River sediments. This study provides valuable insights for ecological restoration and pollution control in the Three Gorges Reservoir area.

## MATERIALS AND METHODS

### Study area and sampling strategy for investigating the physicochemical characteristics of the Sunxi river sediments

The Sunxi River (106°15′–106°31′E, 28°37′–29°13′N) rises in Jinding Mountain, Guizhou Province, China, flows through Baixi Village, Chongqing, and finally into the Qijiang drainage. As part of the Qijiang River system, it has a total length of 120 km, a drainage area of 1,198 km$^2$, and an annual average flow of 19.9 m$^3$/s. The vegetation type is evergreen broad-leaved forest, with the main typical tree species being Masson pine (*Pinus massoniana* Lamb.), Chinese fir (*Cunninghamia lanceolata*), and cypress (*Cupressus funebris* Endl.). The annual average rainfall ranges from 1,000–1,450 mm, with precipitation from May to September accounting for 70%–75% of the annual total. A subtropical monsoon humid climate predominates, with an annual average temperature of 13.6 °C–18.3 °C and rain heat synchronization. The depth of the Sunxi River varies depending on the river section, season, and hydrological conditions. The riverbanks are generally 10–20 m high, and the river width ranges from 5–50 m. The river serves as an essential waterway for transportation and resource flow in Chongqing. Its perennial navigable section, extending over 30 km, supports material transport, water storage and ecological functions, while also contributing to the cultural and landscape value of the region.

The sampling points were arranged every 3–4 km starting from Simian Mountain, with a total of 53 sampling points along the river in May 2021. Field surveys were carried out with topographic factors (elevation, slope, slope aspect, slope position), and the habitat data are presented in Table 1. Cluster analysis was performed with altitude, slope and aspect as environmental factors *via* R software, and the Sunxi River was finally divided into three sections: upper, middle and lower reaches. Sediment samples were collected within 10 cm of the top surface sediments with a Peterson grab sampler, and water was sampled 0.5 m below the surface. In total, 53 water samples and 34 sediment samples were collected according to the field conditions, and the sampling sites were mapped *via* ArcGIS 10.8 (Fig. 1). The air-dried sediments were finely ground and sieved to pass through two mm

**Table 1  Characteristics of the Sunxi River.**

| Reach | | Upper reach | Middle reach | Lower reach |
|---|---|---|---|---|
| Sample No. | | 1–15, 45–50 | 16–29, 50–53 | 30–44 |
| Elevation(m) | | 364–888 | 189–288 | 162–177 |
| Water quality characteristics | pH | 9.16 ± 0.27 | 8.82 ± 0.33 | 8.75 ± 0.33 |
| | DO(mg/L) | 7.33 ± 0.86 | 7.80 ± 1.19 | 8.42 ± 1.02 |
| | EC(us/cm) | 229.27 ± 50.07 | 258.29 ± 62.67 | 290.31 ± 116.18 |
| | TN(mg/L) | 0.93 ± 0.39 | 0.76 ± 0.25 | 1.16 ± 0.12 |
| | TP(mg/L) | 0.04 ± 0.19 | 0.06 ± 0.04 | 0.06 ± 0.04 |
| Habitat | | exists an enclosed reservoir environment within the Simian Mountain Scenic Area | Numerous orchards cultivating Sichuan pepper and citrus are distributed throughout the area, along with bridges and ferry crossings | many towns and residents in the area, and laundry activities are observed along both sides of the river |

**Notes.**

The data are mean ± standard deviation.

DO, Dissolved oxygen; EC, Electrical conductivity; TN, Total nitrogen; TP, Total phosphorus.

Habitats are based on field surveys.

for the determination of available nutrients and then 0.25 mm for the determination of sediment organic matter (SOM) and total nutrient concentrations.

## Study of nitrogen release from sediment using orthogonal experimental design

The sediment samples with relatively high N pollution levels were selected and mixed as the test material. Our previous single-factor simulation experiments revealed that the overlying water environment factors (temperature, pH, and disturbance) significantly affect the release of N and P from the sediments of the Sunxi River (*Ning et al., 2024*). These findings, along with water quality data, habitat characteristics of the study area, and relevant literature, provide the basis for selecting environmental gradients for orthogonal experiments. The temperature was set to 5 °C, 25 °C, or 35 °C *via* biochemical thermostatic incubators. The gradient included typical winter low temperatures and summer high temperatures, reflecting realistic seasonal variations in the study area (*Hou et al., 2022*). On the basis of the actual pH measurements of the Sunxi River water quality (Table 1), the pH was adjusted to 5.0, 7.0 and 9.0 using 0.5 mol/L HCl and 0.5 mol/L NaOH, aiming to encompass a broad spectrum of aquatic environments, ranging from acidic to neutral to alkaline conditions. This range was particularly relevant given the unique characteristics of the Sunxi River, which typically has a relatively high pH, close to 9.0. By considering these natural conditions within the experimental framework, this study ensures the ecological relevance and applicability of the results to the Sunxi River. Furthermore, extending the pH range to include both acidic (pH 5.0) and neutral (pH 7.0) conditions allows for a comprehensive assessment of pH-related effects across different ecological scenarios (*Lu et al., 2022*). Hydraulic disturbances caused by varied water flows and wind-generated waves were simulated by reciprocating a water-bath constant-temperature oscillator (two

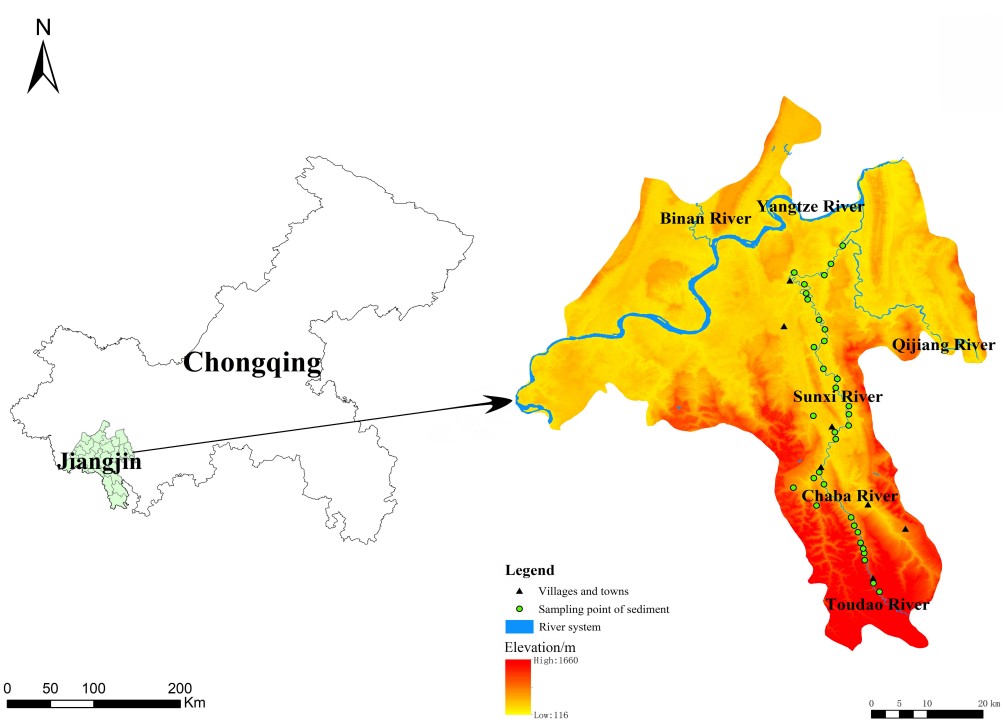

**Figure 1  Sediment sampling sites in the Sunxi River.**

disturbance speeds were set, including a medium speed of 100 r/min and a high speed of 200 r/min).

$L_9(3^4)$ refers to a specific type of orthogonal array in an experimental design. $L_9$ indicates that the design matrix consists of 9 rows, each representing a unique combination of factor levels to be tested; $(3^4)$ signifies that there are at most four factors (the superscript "4"), and each factor has three levels (the base "3"). The $L_9(3^4)$ orthogonal array allows for the efficient study of four factors at three levels each, using only nine experimental runs (*Hunter, 1989*). This design is particularly useful in situations where conducting a full factorial experiment (which would require $3^3=27$ runs) is impractical because of constraints such as time, cost, or resources. Instead, the $L_9$ array provides a balanced representation of all factors and their levels, enabling the analysis of main effects and some interaction effects without the need for a full factorial design. This design is a powerful tool in experimental research for optimizing conditions and analysing multiple variables efficiently. The $L_9(3^4)$ orthogonal experimental design was applied with three environmental factors, three levels each with reference to the historical measured values of the Sunxi River (Table 2).

As shown in Fig. 2, twenty-seven wide-mouth bottles were prepared, each with a capacity of 1 L, 20 g of mixed sediment (created by combining sediment samples collected from multiple locations along the Sunxi River) was laid flat on the bottom of each bottle, and 1 L of distilled water was slowly added as the overlying water. The experiment was kept in a dark place to simulate the real environment of the sediment (this can be important for

**Table 2  Orthogonal experimental design $L_9(3^4)$ for indoor simulation.**

| Treatment | Factors and levels | | | Experimental conditions |
|---|---|---|---|---|
| | A | B | C | |
| | Temperature/ °C | pH | Disturbance/ (r/min) | |
| T1 | 1 (5 °C) | 1 (5) | 1 (0 r/min) | 9 reactors in a refrigerator at 5 °C |
| T2 | 1 (5 °C) | 2 (7) | 2 (100 r/min) | |
| T3 | 1 (5 °C) | 3 (9) | 3 (200 r/min) | |
| T4 | 2 (25 °C) | 1 (5) | 3 (200 r/min) | 9 reactors in a 25 °C incubator |
| T5 | 2 (25 °C) | 2 (7) | 1 (0 r/min) | |
| T6 | 2 (25 °C) | 3 (9) | 2 (100 r/min) | |
| T7 | 3 (35 °C) | 1 (5) | 2 (100 r/min) | 9 reactors in a 35 °C incubator |
| T8 | 3 (35 °C) | 2 (7) | 3 (200 r/min) | |
| T9 | 3 (35 °C) | 3 (9) | 1 (0 r/min) | |

Notes.

T1–T9 were the treatments of the $L_9(3^4)$ orthogonal experimental design, and the distribution of rows and columns of the $L_9(3^4)$ orthogonal experimental design was according to *Lu et al. (2022)*.

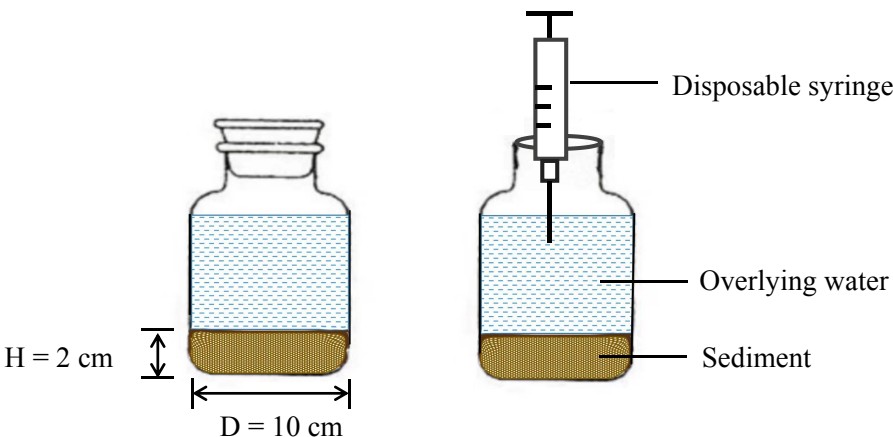

**Figure 2  Schematic diagram of the reactor.**

studying processes unaffected by light, such as nonphotosynthetic microbial activity or the stability of light-sensitive chemicals). During incubation, the bottles were tightly corked and sealed, and 130 mL of overlying water was extracted with a disposable syringe when water samples were collected on Days 1, 2, 3, 5, 7, 9, 13 and 17 (*Peng et al., 2021*). The same volume of distilled water was added along the inner wall of the reactor to keep the volume of overlying water constant (to avoid disturbing the sediment during the water addition process). All the experiments were repeated three times, resulting in a total of 216 water samples. The orthogonal experimental design aims to simulate close-to-natural conditions while maintaining controlled variables to isolate specific effects. This approach is typical in

environmental science, where it is crucial to understand the impact of individual factors in complex ecosystems.

The N release intensity and release rate from sediments can be calculated as follows:

$$R_i = [V(c_n - c_0) + \sum_{j=1}^{n} V_{j-1}(c_{j-1} - c_a)]/S \qquad (1)$$

$$R_i' = R_i/t \qquad (2)$$

$V$ and $V_{j-1}$ (L) are the volumes of the overlying water and sampling water, respectively; $c_0$, $c_n$, $c_{j-1}$ and $c_a$ (mg/L) are the initial pollutant concentration, pollutant concentration on day $n$ and day $j-1$, respectively, and that of compensating sampled water; $S$ (m$^2$) is the surface area of the sediment–water interface; and $t$ (day) is the incubation time. $R_i$ (mg/m$^2$) is the N release intensity on the $i$th day, which is indicative of how much N is released from the sediment into the overlying water per unit area of sediment. $R_i'$ (mg/(m$^2$·d)) is the pollutant flux until the $i$th day, and it represents the rate at which the pollutant is transferred across the sediment−water interface until a specific day (day '$i$'). $R_i' > 0$ means that the pollutant spreads from the sediment to the overlying water; otherwise, it spreads from the overlying water to the sediment. The N release intensity ($R_i$) and N release rate ($R_i'$) are key metrics for understanding nutrient dynamics in aquatic ecosystems and are crucial for assessing the health of these systems, especially in the context of pollution or eutrophication studies.

## Determination of physicochemical properties in the sediments and overlying water

The sediment pH was determined by mixing 10 g of sediment with 25 mL of deionized water, and the sediment electrical conductivity (SEC) was measured with a conductivity meter. The sediment available phosphorus (SAP) was extracted from sediment samples with 0.03 mol/L NH$_4$F–0.025 mol/L HCl and measured at 700 nm *via* a spectrophotometer. The readily available potassium (SAK) was extracted from sediments with 1 mol/L NH$_4$OAc (pH=7.0) and measured *via* a flame spectrophotometer (*Tian et al., 2016*). The SOM content was determined *via* the wet combustion method (*Nelson & Sommers, 1996*). In accordance with *Kavvadias et al. (2001)*, the concentrations of sediment total nitrogen (STN) and total phosphorus (STP) were determined *via* sulfuric acid-hydrogen peroxide digestion, followed by Kjeldahl analysis and the Mo−Sb colorimetric method, respectively.

Deionized water was used to prepare the solution and to rinse each container. The concentrations of TN, NH$_4$-N and NO$_3$-N in the water samples were determined *via* a flow injection analyser in accordance with Water and Wastewater Monitoring and Analytical Methods (the Fourth Edition, 2002; *The State Environmental Protection Administration of China (SEPA), 2002*). For laboratory quality control, samples were measured in triplicate, and the precision of the assay was within 5% of the relative standard deviation.

### Data analysis

The experimental data were recorded in Excel 2013 for descriptive statistics. Pearson correlation analysis and regression analysis were performed *via* SPSS 24 to examine the

**Table 3  Descriptive statistics of physicochemical properties of sediments in the Sunxi River ($n = 34$).**

| Indicator | Minimum | Maximum | Mean | Standard deviation | Skewness | Kurtosis | Coefficient of variation (%) |
|---|---|---|---|---|---|---|---|
| pH | 5.20 | 7.67 | 6.74 | 0.42 | −0.54 | 5.70 | 6 |
| SEC (us/cm) | 34.50 | 140.90 | 65.57 | 24.63 | 1.35 | 1.89 | 38 |
| SOM (g/kg) | 0.73 | 46.31 | 10.04 | 10.29 | 1.93 | 4.03 | 102 |
| STN (mg/kg) | 40 | 2,130 | 430 | 0.45 | 2.08 | 5.20 | 107 |
| STP (mg/kg) | 110 | 480 | 200 | 0.09 | 1.22 | 1.59 | 42 |
| SAP (mg/kg) | 6.82 | 56.73 | 15.99 | 8.81 | 3.16 | 13.69 | 55 |
| SAK (mg/kg) | 31.08 | 134.66 | 61.24 | 24.38 | 1.05 | 1.09 | 40 |

Notes.

SEC, sediment electrical conductivity; SOM, sediment organic matter; STN, sediment total nitrogen; STP, sediment total phosphorus; SAP, sediment available phosphorus; SAK, sediment readily available potassium; $n = 34$, The same below.

linear relationships between the physicochemical properties of sediments. The correlation coefficients were calculated, and statistical significance was assessed at $p < 0.05$. A line chart of N release intensity in different forms was generated *via* GraphPad Prism 9. The concentrations of all the N forms used in the test are presented as the means±standard deviations (SDs). To compare the effects of different environmental factors on N release, we applied one-way analysis of variance (ANOVA) to determine whether there were statistically significant differences between the means of the groups under varying conditions at a significance level of $p < 0.05$ *via* MATLAB 21.0. *Post-hoc* tests were conducted to identify which specific conditions led to significant differences.

## RESULTS

### Sediment characteristics of the Sunxi River

The sediment pH ranged from 5.2 to 7.7 and was neutral or slightly acidic overall (Table 3). As an important indicator of the degree of organic nutrients in the sediment, the minimum, maximum and mean SOM contents were 0.73 g/kg, 46.31 g/kg and 10.04 g/kg, respectively, with large coefficients of variation, indicating high variability. The maximum STN and STP were 2,130 mg/kg and 480 mg/kg, respectively, and the mean STN and STP were 430 mg/kg and 200 mg/kg, respectively. The concentration of STN was significantly greater than that of STP, with a greater coefficient of variation. The STP, SAP and SAK all exhibited moderate variability. The mean values of SAP and SAK were 15.99 mg/kg and 61.24 mg/kg, respectively. The kurtosis of all monitoring indicators was greater than 0, indicating a leptokurtic distribution. With the exception of pH, all the other indicators presented positive skewness.

There were correlations among the physicochemical properties of the sediments in the Sunxi River Basin (Fig. 3). The concentration of STN was highly significantly positively correlated with the concentrations of SOM, STP, SAP, and SAK ($P < 0.001$), whereas its correlation with pH was relatively weak and significant ($P < 0.05$). The STN content was positively correlated with the SOM content, and the linear equation was *[STN] =*

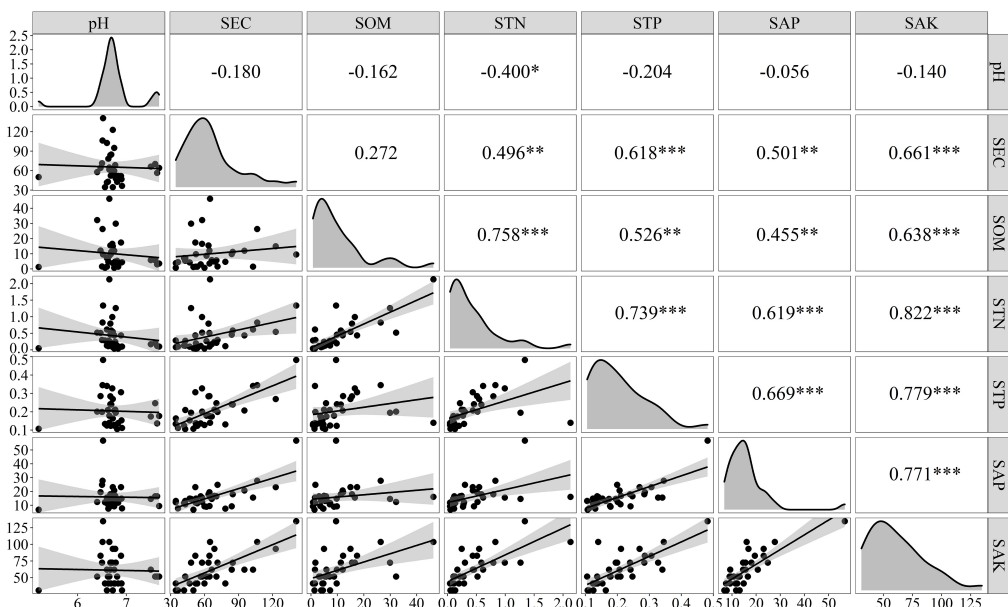

**Figure 3 Correlation and regression analysis of the physicochemical properties of the sediments.**
Note: An asterisk (*) indicates a significant correlation at $p < 0.05$, ** indicates a very significant correlation at $p < 0.01$, and *** indicates a very significant correlation at $p < 0.001$.

$0.062[SOM] + 0.067$ ($R^2 = 0.658$, $P < 0.01$, $n = 34$), and the STN concentration increased with increasing SOM, indicating strong synergistic effects.

## Release of N from sediments under different environmental factors

On the basis of the experimental data, the release intensity and release rate of different forms of N were calculated according to formulae (1) and (2) in the 'Materials and Methods' section. As shown in Fig. 4A, except for treatments T3 and T4, the release intensity of TN gradually increased with time, peaking at the final sampling. The highest release intensity occurred at T8 (temperature = 35 °C, pH = 7, disturbance = 200 r/min). High release intensities were observed in the treatments at 35 °C (T7, T8, and T9), which were significantly greater than those in the other six treatments, with rapid increases on days 1–2, 3–5 and 9–13. The release trend of $NH_4$-N was similar to that of TN, and the maximum release intensity was reached at 35 °C (Fig. 4B). The release intensity at high temperature (35 °C) was more than twice that at low temperature (5 °C) on the first day. Unlike TN, the release intensity of $NH_4$-N significantly decreased after the 9th day at 35 °C. The maximum release intensity of $NO_3$-N occurred on the last day of the simulation experiment, and the release intensities of T7, T8, and T9 increased from days 13–17, with T8 (pH = 7, disturbance = 200 r/min) > T7 (pH = 5, disturbance = 100 r/min) > T9 (pH = 9, disturbance = 0 r/min), indicating that the release rate of $NO_3$-N was lower than that of the other forms of N.

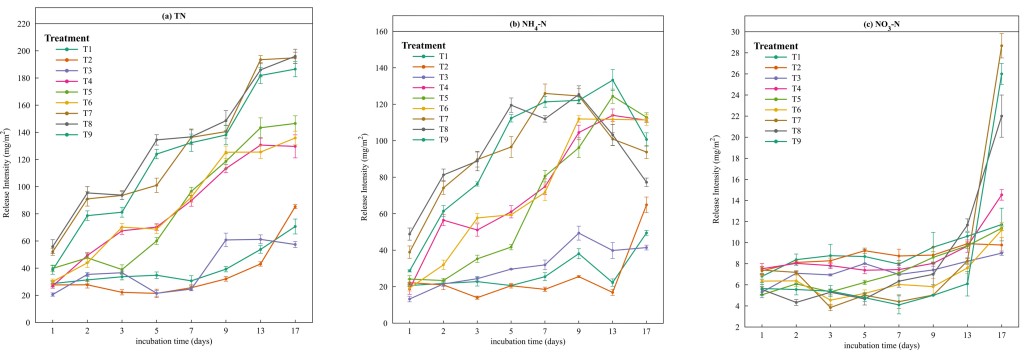

**Figure 4  Line chart of the release intensity of different forms of nitrogen.**

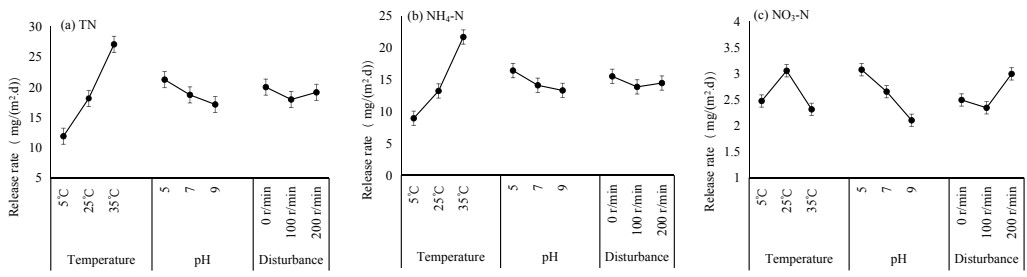

**Figure 5  Main effect plots of the N released from the Sunxi River sediments during the orthogonal experiments.**

A comparison of the main effect plots of the orthogonal tests (Fig. 5) revealed that the TN release rate increased with increasing temperature, reaching a maximum at 35 °C. Similarly, the $NH_4$-N release rate increased with increasing temperature and decreased with increasing water pH, and its release rate reached a maximum at 35 °C. Among the three environmental factors, temperature was the most significant factor affecting TN and $NH_4$-N release. The release rate of $NO_3$-N decreased with increasing water pH, with the highest release rate occurring at pH 5, and the maximum release rate occurred under disturbance conditions of 200 r/min; thus, pH was the most significant factor affecting $NO_3$-N release. The ANOVA confirms that temperature had a significant effect on TN and $NH_4$-N, but not on $NO_3$-N, whereas pH and disturbance had no statistically significant effects (Table 4).

## Comparison of the effects of significant levels of environmental factors on N release from sediments

The maximum mean release rates of TN, $NH_4$-N and $NO_3$-N were 29.24 mg/(m²·d), 23.11 mg/(m²·d), and 4.32 mg/(m²·d), respectively (Table 5), and the maximum values all occurred in T8. To further analyse the order in which environmental factors affect the release rate of N from sediments, a range analysis was conducted on the experimental data of TN, $NH_4$-N, and $NO_3$-N release. The ranges of TN were 15.16, 4.09 and 2.04

**Table 4 Analysis of variance and significance test of each factor in simulation experiment of N release from the sediments.**

| Nitrogen form | Environmental factor | Sum of square | Degree of freedom | Mean square | F | Significance |
|---|---|---|---|---|---|---|
| | Temperature | 348.472 | 2 | 174.236 | 13.378 | 0.007[**] |
| TN | pH | 25.581 | 2 | 12.791 | 0.982 | 0.505 |
| | Disturbance | 6.298 | 2 | 3.149 | 0.242 | 0.805 |
| | Temperature | 252.707 | 2 | 126.354 | 13.081 | 0.023[*] |
| $NH_4$-N | pH | 15.727 | 2 | 7.863 | 0.814 | 0.551 |
| | Disturbance | 4.173 | 2 | 2.087 | 0.216 | 0.822 |
| | Temperature | 0.904 | 2 | 0.452 | 1.089 | 0.479 |
| $NO_3$-N | pH | 1.43 | 2 | 0.715 | 1.723 | 0.367 |
| | Disturbance | 0.703 | 2 | 0.352 | 0.848 | 0.541 |

**Notes.**
[*]A significant effect at $p < 0.05$.
[**]A very significant effect at $p < 0.01$.

**Table 5 Orthogonal range analysis of sediments in the Sunxi River.**

| Treatment | A | B | C | Mean release rate (mg/(m²·d)) | | |
|---|---|---|---|---|---|---|
| | Temperature | pH | Disturbance | TN | $NH_4$-N | $NO_3$-N |
| T1 | 1 (5 °C) | 1 (5) | 1 (0 r/min) | 17.45 | 13.72 | 2.67 |
| T2 | 1 (5 °C) | 2 (7) | 2 (100 r/min) | 9.21 | 6.58 | 2.67 |
| T3 | 1 (5 °C) | 3 (9) | 3 (200 r/min) | 8.97 | 6.52 | 2.08 |
| T4 | 2 (25 °C) | 1 (5) | 3 (200 r/min) | 19.16 | 13.77 | 2.58 |
| T5 | 2 (25 °C) | 2 (7) | 1 (0 r/min) | 17.64 | 12.67 | 2.71 |
| T6 | 2 (25 °C) | 3 (9) | 2 (100 r/min) | 17.56 | 13.24 | 2.11 |
| T7 | 3 (35 °C) | 1 (5) | 2 (100 r/min) | 27.04 | 21.80 | 2.23 |
| T8 | 3 (35 °C) | 2 (7) | 3 (200 r/min) | 29.24 | 23.11 | 4.32 |
| T9 | 3 (35 °C) | 3 (9) | 1 (0 r/min) | 24.84 | 20.17 | 2.11 |
| Range (TN) | 15.16 | 4.09 | 2.04 | | | |
| Range ($NH_4$-N) | 12.75 | 3.12 | 1.65 | | | |
| Range ($NO_3$-N) | 0.74 | 0.97 | 0.66 | | | |
| Significance ranking | | | | A>B>C | A>B>C | B>A>C |

for temperature (T), pH and disturbance level (D), respectively. Within the specified range of environmental conditions, the range (T) > range (pH) > range (D). Therefore, the significance ranking of factors influencing TN release was T >pH >D. Similarly, the ranges of $NH_4$-N were 12.75, 3.12 and 1.65, and those of $NO_3$-N were 0.74, 0.97 and 0.66, respectively, for T, pH and D. Range analyses revealed that the significance ranking of factors influencing the release of $NH_4$-N was T >pH >D, and for the release of $NO_3$-N, the significance ranking was pH >T >D.

## Establishment of multiple linear regression equations for sediment N release

Multiple linear regression analysis is applied to explore the relationship between a dependent variable $y$ and two or more general independent variables $x_1$, $x_2$, ..., and

**Table 6 Parameters of multiple linear regression empirical equations for nitrogen in the sediment.**

| N form | $b_0$ | $b_1$ | $b_2$ | $b_3$ | Coefficient of determination $R^2$ | Adjusted $R^2$ | P |
|--------|-------|-------|-------|-------|-----------------------------------|----------------|---|
| TN | 16.249 | 0.478 | −1.023 | −0.004 | 0.548 | 0.526 | 0.00[**] |

Notes.
[**]A very significant effect at $p < 0.01$.

$x_k$. The basic expression is $y = b_0 + b_1 x_1 + b_2 x_2 + ... + b_k x_k + \varepsilon$, where $b_1, b_2, ..., b_k$ represents the corresponding regression coefficient, $b_0$ is a constant and $\varepsilon$ is the error (*Zhu et al., 2023*).

Multiple linear regression equations were established between different forms of N and the three environmental factors of temperature, pH and disturbance level. As shown in Table 6, the multiple linear regression model for TN had moderate explanatory power, with $R^2 = 0.548$ and adjusted $R^2 = 0.526$. However, the models for $NH_4$-N and $NO_3$-N had $R^2$ and adjusted $R^2$ values of less than 0.2, indicating insufficient explanatory power. Furthermore, the regression equations for $NH_4$-N and $NO_3$-N were statistically nonsignificant. Therefore, only the regression equation for TN was reliable for interpreting N release dynamics in the sediment.

After the parameters in Table 6 were substituted into the equations, the release rate $R'$ (mg/(m²·d)) was calculated as follows: $R'_{TN} = 0.478 * T − 1.023 * pH − 0.004 * D + 16.249$.

# DISCUSSION

## Characterization of sediment pollution in the Sunxi River

The sampling points in this study were distributed across the Chongqing section of the Sunxi River, covering areas with significant altitude variation and substantial human interference. Therefore, the coefficients of variation of STN and SOM concentrations are large and have obvious regional variation characteristics. The STN concentration ranged from 40–2,130 mg/kg, and the STP concentration ranged from 110–480 mg/kg. There were correlations among the physicochemical indicators of the sediments. The STN and SOM exhibited a highly significant positive correlation ($P < 0.01$), indicating a strong synergistic relationship. This finding was similar to the results of *Hou et al. (2022)* in the Sunxi River, suggesting that N pollution was more intense than P pollution. The enrichment of organic matter in the sediment was the main source of STN. The mineralization of organic matter in sediments is closely related to N sources and sedimentary processes. At the same time, a significant positive correlation was observed between STN and STP ($P < 0.01$), suggesting shared pollution sources or depositional environments for both elements.

Compared with more isolated reservoir environments, the Sunxi River watershed contributes significant terrestrial organic carbon to the reservoir through the river confluence, increasing the risk of nutrient release from surface sediments. The average SOM concentration in the sediments of the Sunxi River was 10.04 g/kg, with a predominantly loamy texture and high surface adsorption capacity. SOM plays a dual role: it aids in natural water purification by adsorbing and breaking down pollutants and excess nutrients, but under certain conditions, the enriched nutrients can be rereleased into the water, triggering

algae blooms and disrupting the aquatic ecosystem's balance. Most SOM originates from sewage discharge, and the long-term accumulation of aquatic organisms remains. SOM mineralization occurs in the sediment surface layer by consuming significant amounts of DO and thus drives nutrient salt release and migration. As phytoplankton increase and microorganisms die, SOM input and accumulation intensify, creating a vicious cycle (*Fan et al., 2019*). In the middle and lower reaches of the Sunxi River watershed, elevated STP levels are attributed primarily to frequent anthropogenic activities. These include extensive fertilizer use in peppers and citrus plantations, household laundry along riverbanks, and the direct discharge of untreated sewage and animal waste into the river.

**Effects of environmental factors on N release from sediments in the Sunxi River**

In riverine reservoirs, temperature affects water bodies predominantly through hydrothermal stratification, disrupting vertical nutrient movement (*Huang et al., 2019*). When nutrients are abundant, relatively high temperatures accelerate phytoplankton growth, often causing hypoxia due to excessive algae proliferation. In Chongqing, summer temperatures can reach 40 °C, promoting rapid algae blooms, increasing microbial activity, and reducing sediment oxygen levels, thereby increasing N release. Therefore, N pollution peaks in summer, which is consistent with the findings of *Malnik et al. (2022)*. At lower temperatures, elevated DO concentrations in the overlying water are beneficial for aerobic processes, including nitrification. Certain species of nitrifying bacteria convert some $NH_4$-N to $NO_3$-N, resulting in an increase in its concentration, which is consistent with the results of *Nguyen et al. (2019)* and *Nweze & Eze (2018)*. In general, pH affects N release by influencing the activity of microorganisms in the sediment, which is a slow, long-term process, and most microorganisms are more active in a neutral environment (pH = 6.5–7.5). Since the experiment lasted 18 days, pH primarily affects the variable charge on sediment colloid surfaces (*Chen et al., 2019*). Neutral sediments resulted in greater connectivity and stability in diazotrophic communities, increasing N fixation, whereas acidic or alkaline conditions reduced the stability and efficiency of these processes (*Luo et al., 2021*). The sediment fixation of exchangeable N was greater under neutral conditions, whereas both acidic and alkaline conditions promoted N release. This is consistent with *Lei et al. (2016)* in Jiangjin city, Chongqing, where an increased $H^+$ concentration under acidic conditions lowered the pH. In neutral environments, relatively high microbial activity promotes $NO_3$-N conversion, making $NO_3$-N more readily available for microbial uptake and less prone to environmental release (*Li et al., 2020*; *Yao et al., 2018*). Hydrodynamic disturbances are vitally important for the release of endogenous nutrients in lakes and reservoirs. A decrease in water clarity, along with a significant increase in turbidity and mineralization, adversely affects the water ecosystem by hindering the growth and reproduction of aquatic organisms while also compromising water quality and ecological health (*Zhao et al., 2018*). Disturbance resuspends particulate matter in sediments and increases the exchange area between interstitial water and overlying water, thereby facilitating nutrient diffusion (*Zhang et al., 2019*). Greater disturbance intensity enhances nutrient release at the sediment–water interface. In the Three Gorges Reservoir, water level scheduling creates alternating wet–dry

cycles, resulting in unique hydrodynamic conditions in the Sunxi River and reducing the thickness of the diffusion boundary layer. To protect a river's hydrological environment, efforts should focus on raising environmental awareness, minimizing anthropogenic disturbances, and improving gate and dam management while controlling external nutrient inputs.

The intensity and rate of N release from sediments are influenced by multiple factors, including SOM, water temperature, depth, DO, pH and water disturbance, which interact to affect the release process. The release intensity of each factor generally increased over time, peaking at the final sampling. This is in agreement with the findings of *Zhang (2019)*, who reported that N release from Zhangze Reservoir sediments increased with temperature and followed a similar pattern at the single-factor level. Range analysis revealed that the environmental factors influenced TN and $NH_4$-N release from the Sunxi River in decreasing order of significance: temperature >pH >disturbance. This pattern is consistent with the results of *Wu et al. (2014)* for Xuanwu Lake, China. Although pH did not have a statistically significant effect on $NO_3$-N release, the results from Tables 4 and 5 suggest that pH had the greatest overall influence among the three environmental factors, which aligns with established knowledge on the sensitivity of nitrate to pH variations in aquatic environments. Similar to the results of *Peng et al. (2021)*, the significance ranking of $NO_3$-N release was pH >temperature >disturbance. Under weakly acidic conditions, the release rate of $NH_4$-N decreased, whereas the release rate of $NO_3$-N increased. In a more alkaline environment, the release rate of $NH_4$-N increased, whereas the release rate of $NO_3$-N decreased (*He et al., 2015*). The effects of these three environmental factors on sediment N release interact with extremely complex processes. Moreover, in actual river systems, interference from other factors, such as water flow velocity, nutrient content, and microbial communities, may occur.

The models for $NH_4$-N and $NO_3$-N exhibited very low $R^2$ and adjusted $R^2$ values (0.236 and 0.222, respectively) and were statistically nonsignificant. This suggests that the release rates of $NH_4$-N and $NO_3$-N in the sediment may be influenced by other key environmental factors, such as organic matter content, redox potential, and microbial activity. Our results limit the interpretation of $NH_4$-N and $NO_3$-N dynamics under the current experimental conditions but highlight the need for further investigations with additional variables to better understand the controlling factors. A comprehensive understanding of N release from sediments requires to the consideration of multiple interacting factors. Given the vulnerability of the sediment–water interface to environmental changes, effective pollution mitigation relies on technologies tailored to the specific sediment characteristics of each reservoir. These characteristics must be carefully and continuously evaluated to ensure that management strategies remain aligned with the reservoir's unique environmental conditions (*Carter et al., 2003*).

## CONCLUSIONS

This study used orthogonal simulation experiments to examin the effects of temperature, pH, and hydraulic disturbance on N release from Sunxi River sediments. The results

revealed that the concentration of STN was significantly greater than that of STP from Sunxi River sediments, STN was highly significantly positively correlated with SOM, and temperature, pH, and disturbance all have different degrees of influence on the process of N release from sediments, with temperature playing the most important role in the behavior of TN and $NH_4$-N release from sediments. The maximum N release intensities occurred at high temperature (35 °C), neutral pH (7.0), and strong disturbance (200 r/min). Additionally, a significant positive correlation was observed between STN and SOM. These findings provide insights for water quality management and pollution control in the Sunxi River and similar aquatic systems. It is recommended that future studies explore the interactions between temperature and microbial activity in more detail, as these studies could provide further insights into the mechanisms driving N release. Additionally, the practical implications of the results suggest that managing temperature and pH in aquatic systems could be a viable strategy for mitigating N pollution, particularly in areas prone to eutrophication.

## ACKNOWLEDGEMENTS

We would like to thank the residents within the Sunxi River watershed for their hospitality during our fieldwork.

### Funding

This research was funded by National Key R&D Program of China (Grant No. 2017YF0505306). The funders had no role in study design, data collection and analysis, decision to publish, or preparation of the manuscript.

### Grant Disclosures

The following grant information was disclosed by the authors:
National Key R&D Program of China: 2017YF0505306.

### Competing Interests

The authors declare there are no competing interests.

### Author Contributions

- Yihong Ning conceived and designed the experiments, performed the experiments, analyzed the data, prepared figures and/or tables, authored or reviewed drafts of the article, and approved the final draft.
- Bin Gao conceived and designed the experiments, authored or reviewed drafts of the article, and approved the final draft.
- Haiyan Wang conceived and designed the experiments, authored or reviewed drafts of the article, and approved the final draft.
- Wenning Hou performed the experiments, authored or reviewed drafts of the article, and approved the final draft.

## Data Availability

The raw measurements are available in the Supplementary File.

## Supplemental Information

Supplemental information for this article can be found online at http://dx.doi.org/10.7717/peerj.19161#supplemental-information.

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
