# Peer review of "Effects of pH, temperature and hydraulic disturbance on nitrogen release from sediments in the Sunxi River, Three Gorges Reservoir Area, China"

_PeerJ, doi:10.7717/peerj.19161_

## Round 0.1 · original submission · Major Revisions

According to the reviewers: The authors should address all the concerns, including making the text more readable and improving the English.

Reviewer 1 ·

Basic reporting

I would like to thank the authors for this contribution dealing with the study of nitrogen release from sediment in highly contaminated area, which are prone to eutrophication phenomena.

Authors are strongly advised to improve the English language of the document. There are many long sentences where the meaning is not obvious, making the manuscript, sometimes, difficult to read. Some examples where the English should be revised are in: lines 44-46, lines 67-71, lines 81-82, lines 109-110, lines 116-119, lines 121-122, lines 185-188, lines 309-310, lines 310-314, lines 331-332, lines 367-369, lines 373-374, lines 382-384, lines 390-395, lines 413-415, lines 417-419, lines 421-424, lines 428-431, lines 435-439, lines 444-445, lines 445-450, lines 451-453. Also, technically I prefer to use TN, NH4-N, NO3-N in the whole document to express the different forms of nitrogen rather than total, ammonia and nitrate N.
The authors present an in-depth overview of the subject, with relevant references and a better focus on the issue. However, I think the introduction section is too long and could be shortened a little.
The article follows a standard structural format beginning with the abstract, introduction, material and methods, discussion and conclusion. But nowhere have I found the keywords for the abstract!
I also have an issue of confusion between two sections ‘Orthogonal Experimental Design’ and ‘Statistical Analysis’ in Materials and Methods, since Orthogonal Design is part of Statistical Analysis. I suggest to the authors (if they agree) to keep only the ‘Orthogonal Experimental Design’ section and insert each explanation given in ‘Statistical Analysis’ in its appropriate section. For example, ‘Sediment sampling sites... in Excel 2013 for descriptive statistics’ could be placed in the ‘Study area and sampling’ section. Otherwise, authors are strongly encouraged to find an alternative, more logical structure between the two sections. Similarly, the section ‘Calculation of nitrogen release from sediments’ should be included in the section ‘Orthogonal experimental design’.
The reference ‘Ning et al., 2024’ is missed in the references List.
The resolution of Figure 3 should be improved.

Experimental design

By reading the article, we easily understand that the authors have set up a simulation experiment to determine the most important factors responsible for the release of TN, NH4-N and NO3-N from sediments, and this is the main objective of the study. Nevertheless, the authors should justify which mechanisms and processes have been revealed (last sentence of the introduction) only by studying the influence of the above-mentioned factors or reformulate the objective.
The authors described a L9(3^4) design (with 9 experiments, three levels and four factors), in accordance with Lu et al. (2022), but built an array with 9 experiments and three factors (see lines 203-215). Please provide an explanation regarding this point.
In ‘Statistical analysis’ section, please give more details about the use of the Pearson correlation matrix and the ANOVA test (see lines 273-276).

Validity of the findings

In lines 357-363, the authors have given the equation for nitrogen release without any comment on the result of table 6. If we look at Table 6, the R^2 and adjusted R^2 are very low for NO3-N and NH4-N and medium for TN, which may indicate that the multiple linear regression model is not appropriate to explain the variability of the factors. The authors should provide 1) the analysis of variance of the model, 2) the analysis of variance of the coefficients and 3) the analysis of variance of the residuals, and check whether the multiple linear regression model is suitable or not.
In lines 453-454, the authors state that pH is the more significant factor affecting NO3-N release from sediment. However, results from table 4 shows that only temperature have a significant effect on TN and NH4-N release from sediment. Please revise this in the document.
The conclusion should be rewritten; it does not summarise the results sufficiently. In addition, unnecessary information is given from line 479 to the end.

Reviewer 2 ·

Basic reporting

The introduction is well-written but needs to be split into multiple paragraphs for readability purposes. Lines 74-108 could be split into three paragraphs, focusing on how pH, hydraulic disturbance, and temperature affect N. This would make it easier for readers to understand the logic.

Line 162. Please check the degree signs; they seem unclear.

The sampling time is not mentioned, as dry and wet seasons could influence Nitrogen mobilization between sediments and water. The depth of the river is also not mentioned. Can this river also be affected by land use?


Line 163. I understand what the authors are trying to convey by saying "traffic artery," but this is unscientific writing. Please rephrase it.

Figure 3 is quite hard to read.

Experimental design

Line 216-217. Please confirm if it was dry or wet sediment. Also, the particle size influences nutrient movement, so that's unclear.

If you are mimicking a close-to-real environment, would adding a water sample from the river do that since you have already collected it? Please explain why that was not done since distilled water can potentially dilute the concentrations. was dilution accounted for in your calculations?

If only 1L of distilled water was added, how did you perform eight extractions?

Lines 216-229. Please add a schematic diagram or a figure from the experiment to help the readers better understand.

Please provide details on how the mixed sediment was created.

Validity of the findings

no comment

Additional comments

no comment

---

## Round 0.2 · Minor Revisions

According to the reviewers, your manuscript has been improved. However, some minor changes are needed before it be acceptable for publication

Reviewer 1 ·

Basic reporting

The manuscript is now significantly clearer and more coherent. However, the quality of the English throughout the document still requires improvement, and the authors are strongly encouraged to seek professional language assistance to address this.

Experimental design

- I suggest revising the title "Study Area and Sampling" in the Materials and Methods section to "Study area and sampling strategy for investigating the physicochemical characteristics of Sunxi river sediments" to provide greater specificity and clarity.

- The title "Orthogonal Experimental Design" does not fully capture the scope of this section, which begins with experiments on nitrogen release from sediment and proceeds to the application of this design. I suggest revising the title to: "Study of Nitrogen Release from Sediment Using Orthogonal Experimental Design" for better alignment with the content.

- Lines 260–262: Please revise the phrase, as the Pearson correlation matrix in your paper is used to examine the linear relationships between the physicochemical properties of sediments. Accordingly, the sentence in lines 263–264 should be removed

Validity of the findings

- Line 323: "The ANOVA confirmed these findings, showing that temperature significantly affected N release from sediments". In your case, the ANOVA confirms that temperature had a significant effect on NT and NH4-N, but not on NO3-N. Please revise this accordingly.
- Lines 348-351: please add a reference
- Line 411: please add a reference.
- Line 453: Referring to Table 6 is unnecessary, as the table only presents parameters for TN release from sediment. Please remove this.

Conclusion: Your work devotes an important section to the physicochemical properties of sediments in the Sunxi River. This should be highlighted as one of the main objectives of the study, and key results should be highlighted.

Reviewer 2 ·

Basic reporting

The authors have implemented and rebutted my comments effectively.

Experimental design

I agree with their rationale on their experimental design.

Validity of the findings

No problems here.

---

## Round 0.3 · accepted · Accept

The authors have addressed the reviewers' comments.